# Roles and Mechanisms of NLRP3 in Influenza Viral Infection

**DOI:** 10.3390/v15061339

**Published:** 2023-06-08

**Authors:** Junling Niu, Guangxun Meng

**Affiliations:** The Center for Microbes, Development and Health, CAS Key Laboratory of Molecular Virology & Immunology, University of Chinese Academy of Sciences, 320 Yueyang Road, Life Science Research Building B-205, Shanghai 200031, China; jlniu@ips.ac.cn

**Keywords:** influenza virus, pattern recognition receptor, NLRP3 inflammasome, cytokine, cell death, commensal microbiota

## Abstract

Pathogenic viral infection represents a major challenge to human health. Due to the vast mucosal surface of respiratory tract exposed to the environment, host defense against influenza viruses has perpetually been a considerable challenge. Inflammasomes serve as vital components of the host innate immune system and play a crucial role in responding to viral infections. To cope with influenza viral infection, the host employs inflammasomes and symbiotic microbiota to confer effective protection at the mucosal surface in the lungs. This review article aims to summarize the current findings on the function of NACHT, LRR and PYD domains-containing protein 3 (NLRP3) in host response to influenza viral infection involving various mechanisms including the gut–lung crosstalk.

## 1. Introduction

Influenza viruses are responsible for both seasonal epidemics and occasional pandemics, causing considerable health and economic burdens around the world [1,2,3]. Influenza viral infection primarily causes respiratory disease [4,5,6], increases the risk of secondary bacterial infections [7,8], and exacerbates chronic illnesses such as cardiovascular diseases [9,10,11], resulting in severe complications and increased mortality rates [12,13]. Host cells detect influenza viruses and subsequently initiate both innate and adaptive immunity to combat the virus upon infection. Host pattern recognition receptors (PRRs); NACHT, LRR and PYD domains-containing protein 3 (NLRP3) inflammasome; and symbiotic microbiota all play pivotal roles in recognizing and fighting against the influenza virus [14,15,16,17,18,19,20].

## 2. Influenza Virus

### 2.1. Characteristics of the Influenza Virus

Influenza viruses are single-stranded negative-sense RNA (ssRNA) viruses [21] and belong to the Orthomyxoviridae family. They are categorized into types A, B, C, and D based on differences in nucleoprotein (NP) and matrix protein (M) antigenicity [3]. Type A influenza virus poses the severest threat to human health [2]. The influenza A virus (IAV) particle comprises the envelope, core, and matrix proteins. The viral envelope is derived from the host cell membrane and is embedded with hemagglutinin (HA), neuraminidase (NA), and ion channel matrix protein 2 (M2) [22,23]. HA is responsible for viral attachment and entry into host cells [24], while NA facilitates the release of progeny viruses from infected cells [25]. M2 is involved in viral uncoating and plays a role in resistance to antiviral drugs [26,27]. The core consists of the genome, nucleoproteins (NPs), and RNA polymerase complex. The genome contains eight ssRNA segments, and each of the eight ssRNA segments is encapsidated by NPs, forming a ribonucleoprotein (RNP) complex [28]. Matrix protein 1 (M1) interacts with RNPs, providing structural support and facilitating viral assembly [29]. The RNA polymerase complex is composed of polymerase basic protein 1 (PB1), polymerase basic protein 2 (PB2), and polymerase acidic protein (PA), and plays a crucial role in viral transcription, replication, and host adaptation [22,30]. Additionally, the IAV encodes a non-structural protein 1 (NS1), which is essential for the virus to evade host immune responses [31,32,33,34].

### 2.2. Mutations of Influenza A Virus

Based on variations in HA and NA antigenicity, type A influenza viruses are further classified into different subtypes, such as H1N1 and H7N9. In total, 18 HAs and 11 NAs have been identified [35,36,37], which can theoretically form 198 possible subtypes through reassortment events. The genome of each subtype continuously undergoes antigenic drift and shift. Antigenic drift is caused by point mutations in the viral genome, resulting in changes to the surface glycoproteins HA and NA [38]. This process allows the virus to evade host immune responses, consequently reducing vaccine effectiveness [39]. An antigenic shift occurs during reassortment events, leading to the exchange of gene segments between different subtypes of viruses [40]. This exchange results in the emergence of novel viruses with significantly different antigenic properties and may cause new pandemics [41,42]. In addition to HA and NA, mutations in the internal proteins NP and M1 impede the recognition of these proteins by cytotoxic T lymphocytes (CTLs) [43], further facilitating the viral evasion of host cellular immunity [44].

## 3. Recognition of Influenza Virus by PRRs

Innate immune cells employ pattern recognition receptors (PRRs) to detect pathogen-associated molecular patterns (PAMPs) generated during IAV infection, thereby initiating immune responses [17,45]. The PRRs involved in sensing IAV infection include Toll-like receptors (TLRs), retinoic acid-inducible gene-I (RIG-I)-like receptors (RLRs), and NOD-like receptors (NLRs). Toll-like receptor 7 (TLR7) detects the viral single-stranded RNA (ssRNA) in the endosome [17,46], Toll-like receptor 3 (TLR3) senses the viral double-stranded RNA (dsRNA) in the phagosome [17,47], and retinoic acid-inducible gene-I (RIG-I) recognizes 5′-triphosphated viral RNA in the cytosol [17]. Recognition of IAV by these PRRs triggers the production of type I IFN (IFN-I) through the phosphorylation of IFN regulatory factors 3 and 7 (IRF3 and IRF7, respectively) [17]. Through the interferon receptor, IFN-I induces multiple interferon-stimulated genes (ISGs) to control virus replication [48]. The NOD-like receptor NACHT, LRR and PYD domains-containing protein 3 (NLRP3) and the NLRP3 inflammasome have also been found to play a critical role in orchestrating a protective immune response against IAV infection [15,16,49] (Figure 1).

### 3.1. Recognition of the IAV in the Cytoplasm

During IAV infection, the surface glycoprotein haemagglutinins (HAs) of virions bind to host receptors first, and endocytosis then mediates virus entry [22,30]. Following entry, virus replication occurs, viral ribonucleoproteins (vRNPs) and other proteins are synthesized, and progeny virions are assembled for release and spread within the lungs [50]. This process may cause lethality resulting from compromised virus clearance or harmful lung immunopathology [51]. Retinoic acid-inducible gene-I (RIG-I) detects cytosolic 5′-triphosphated viral single-stranded RNA (ssRNA) generated during IAV replication [33]; by subsequently interacting with the mitochondrial antiviral signaling protein (MAVS), it elicits the NF-κB-dependent expression of pro-inflammatory cytokines and the IFN-regulatory factor 3 (IRF3)-dependent production of type I interferon (IFN-I) [17]. In IAV-infected lung epithelial cells, RIG-I recognizes viral RNA and activates the NLRP3 inflammasome directly, as well as through a type I IFN positive feedback loop [52]. In macrophages and dendritic cells, the M2 protein, a proton-selective ion channel produced during IAV replication, triggers NLRP3 inflammasome activation [53] by causing a pH imbalance and the dispersion of the trans-Golgi network (dTGN) that allows NLRP3 to travel to the phospholipid phosphatidylinositol-4-phosphate (PtdIns4P) on dTGN in order to recruit ASC [53,54,55]; the M2 protein also interacts with MAVS, modulating the MAVS-mediated signaling pathway [56]. In addition, influenza A virus infection initiates the assembly of receptor-interacting protein 1 (RIP1) and receptor-interacting protein 3 (RIP3) into a complex [57]. This complex activates dynamin-related protein 1 (DRP1) and promotes its translocation to mitochondria, resulting in mitochondrial damage, reactive oxygen species (ROS) production, and subsequent NLRP3 inflammasome activation [57].

### 3.2. Recognition of the IAV in the Endosome and Phagosome

During infection, the endocytic vesicles carrying IAV particles enter endosomes, where the genomic ssRNAs are released following the acidolysis of IAV particles. These ssRNAs are then detected by TLR7 within the endosomes, stimulating IRF7-dependent IFN-I production, as well as NF-κB-dependent pro-inflammatory cytokine synthesis [17,46]. IAV-infected cells containing viral dsRNAs can be phagocytosed by macrophages. Within these macrophages, the dsRNAs are recognized by TLR3, leading to the induction of NF-κB-dependent pro-inflammatory cytokines and the production of IRF7-dependent IFN-I generation [17,47,58,59]. Furthermore, high-molecular-weight aggregated PB1-F2 fibrils from the IAV are incorporated into the phagolysosomal compartment. Upon acidification, these fibrils activate the NLRP3 inflammasome and induce IL-1β secretion in macrophages [17,60].

### 3.3. Evasion of the IAV from Immune Responses

While hosts attempt to eliminate the IAV, the virus also works diligently to suppress the antiviral immune responses. The envelope proteins HA and NA undergo constant mutation, allowing the IAV to escape neutralizing antibodies. Furthermore, in macrophages, the influenza virus nonstructural protein 1 (NS1) hinders caspase-1 activation through the N-terminal domain of NS1, and consequently inhibits the maturation of IL-1β and IL-18 [31]. NS1 also interacts with RIG-I, and thus impairs the induction of type I interferon by disrupting the activation of transcription factors that drive IFN-β expression [32,33,34]. Meanwhile, PB1-F2 suppresses IFN-I production and NLRP3 inflammasome activation by promoting mitochondrial fragmentation and inducing mitophagy [61,62]. Notably, the highly pathogenic avian influenza virus H5N1 inhibits NF-κB-mediated inflammatory responses in human monocytes through the activation of Rar-related orphan receptor alpha (RORα) [63].

## 4. Functions of NLRP3 Inflammasome in Influenza Virus Infection

### 4.1. Inflammasomes

Inflammasomes, as cytosolic protein complexes, play a critical role in the innate immune system, and provide a rapid response to pathogenic infections and cellular damage [64]. The canonical inflammasomes, including the NLRP1, AIM2, NLRP3, and NLRC4 inflammasomes, are the most extensively studied, and are known for their ability to activate caspase-1 and facilitate the release of IL-1β and IL-18 [65]. The noncanonical inflammasomes, which include the Pyrin inflammasome and the recently discovered caspase-4/5/11-mediated inflammasomes, represent a distinct subgroup with unique activation mechanisms and effector functions [66]. Canonical inflammasomes are multiprotein complexes that are assembled by pattern recognition receptors (PRRs) and adapter protein apoptosis-associated speck-like protein containing a caspase activation and recruitment domain (ASC) and pro-caspase-1 following the detection of pathogenic microorganisms and danger signals in host cells [67,68]. Pattern recognition receptors, including NOD-like receptors (NLRs) and AIM-2-like receptors (ALRs), have been reported to form inflammasomes such as NLRP1, AIM2, NLRP3, NLRC4, NLRP6, NLRP7, and NLRP12 inflammasomes [69,70]. These inflammasomes recognize pathogen-associated molecular patterns (PAMPs) and endogenous danger-associated molecular patterns (DAMPs) to elicit immune responses that protect the host from pathogen challenge, but may also lead to tissue damage [64,65]. The NLRP3 inflammasome, in particular, has been implicated in a variety of inflammatory diseases [15,16,71].

### 4.2. NLRP3 Inflammasome during Influenza Virus Infection

NLRP3 plays a crucial role in both innate and adaptive immunity. In its inactive state, the NLRP3 protein is bound to the ubiquitin ligase-associated protein SGT-1 and heat shock protein 90 (HSP90), maintaining self-inhibition [72]. Upon activation, NLRP3 oligomerizes and serves as a nucleate for the recruitment of the adaptor ASC and pro-caspase-1 to assemble the NLRP3 inflammasome. This leads to caspase-1 activation and the proteolytic maturation and secretion of IL-1β and IL-18 [73,74,75,76,77], as well as the cleavage of gasdermin D (GSDMD), which triggers pyroptosis by forming membrane pores [78,79] and the release of immune factors such as galectin-3 [80]. The NLRP3 inflammasome plays a significant role in host response to influenza virus infection and can be activated by influenzavirus-associated molecular patterns [17,52,53,57,60]. A number of studies have shown that NLRP3 deficiency results in more severe disease [15,16], while rescued or enhanced NLRP3 activity leads to resistance to IAV infection [14,81]. However, NLRP3 inflammasome activation is not always beneficial. In some cases, the excessive activation of the NLRP3 inflammasome may lead to a “cytokine storm” and immunopathology, exacerbating the severity of influenza virus infection [60,82]. This is because the role of NLRP3 inflammasome varies throughout different stages of influenza virus infection [83], as well as when the host encounters various subtypes of the virus [84]. As a result, researchers have begun to explore strategies to modulate NLRP3 inflammasome activity to alleviate infection-induced pathology [83]. For example, the administration of a NLRP3 inhibitor during the early stages (the first five days post infection) of IAV infection can be detrimental, while it proves beneficial in the later stages (the period from the seventh day to the ninth day after infection) [83]. Furthermore, a deficiency in caspase-1 reduces the survival rate of mice during a challenge with the non-lethal H1N1 subtype influenza virus [15,16], but it is protective in a lethal infection with the H7N9 subtype [84]. Moreover, NLRP3 has been identified as a key factor in regulating Th2, Th17, and Treg differentiation [85,86,87], indicating its potential role in adaptive immune responses during influenza virus infection.

#### 4.2.1. The Impact of NLRP3 on Host Survival and Virus Clearance

The roles of the NLRP3 inflammasome in the survival rates of experimental mice during influenza virus infection have been inconsistent. In PR8 (H1N1)-infected mice, wild-type (WT) mice showed a 70% survival rate, while *Nlrp3*^−/−^, *Caspase1*^−/−^, and *Asc*^−/−^ mice demonstrated survival rates of 20% [15] (or 40%) [16], 40%, and 30%, respectively [15,16]. Nigericin, a compound known to enhance NLRP3 activity, was found to be protective for 15-16-month-old BALB/c mice during influenza virus infection [81]. However, one study reported that NLRP3 was not required for protective immunity against influenza virus infection, but ASC, caspase-1, and IL-1R1 were necessary [49]. Research findings on the role of NLRP3 in clearing the influenza virus are also inconsistent. On day 6 or 7 post-infection, Allen et al. found that *Nlrp3*^−/−^ mice had a significant disadvantage in clearing the virus compared to WT mice [15], while Thomas et al. observed no significant difference in viral load between *Nlrp3*^−/−^ and WT mice [16]. In our studies, we found that enhanced NLRP3 inflammasome activity, driven by *Nlrp3*^-R258W^ mutation, increases mouse survival rates and promotes virus clearance via IL-1β-mediated neutrophil recruitment [14]. In addition, we discovered that NLRP3 plays a role in mediating the enhanced production of IFN-I by microbiota-derived acetate [88]. This heightened IFN-I production effectively suppresses influenza virus replication and increases the survival rate of infected mice [88]. Taking these findings into account, it appears that NLRP3 may be more inclined to increase survival rates and aid in virus clearance during H1N1 influenza virus infection. However, given the inconsistencies in the literature, further research is needed to fully understand the role of NLRP3 in response to IAV infection.

#### 4.2.2. Effect of NLRP3 on Cytokine Production and Cell Infiltration

The IL-1β level in the bronchoalveolar lavage fluid (BALF) of *Nlrp3*^−/−^ mice is lower than that in WT mice, while the levels of MIP2, CXCL1, TNFα, and IL-6 are either lower than, or the same as, in WT mice on day 3 after infection [15,16,49]. Allen et al. [15] and Thomas et al. [16] showed significantly reduced infiltration levels of monocytes/macrophages and neutrophils in the lungs of *Nlrp3*^−/−^ and *Caspase1*^−/−^ mice, while Ichinohe et al. [49] showed more neutrophil infiltration in *Nlrp3*^−/−^ than in WT mice. The different results obtained by different research groups could be due to the inconsistency in virus uptake by intranasal infection [89].

#### 4.2.3. NLRP3 and Lung Injury Repair

NLRP3 plays a vital role in repairing lung damage caused by influenza virus infection. On day 3 after infection with the influenza A virus, *Nlrp3*^−/−^ mice displayed localized necrotic bronchiolar epithelial cells and bronchiole blockage by fibrin, neutrophils, macrophages, and necrotic cells, while bronchioles in WT mice remained largely unaffected [16]. Consistently, on day 11 after infection, extensive collagen was deposited in the alveoli and lung interstitium of *Nlrp3*^−/−^ and *Caspase1*^−/−^ mice, whereas only minor collagen deposition occurred in the alveoli of WT mice [16].

#### 4.2.4. The Role of IL-18 and IL-1β in Influenza Virus Infection

As downstream cytokines of inflammasomes, IL-1β and IL-18, have also been investigated for their roles in combating influenza virus infection. During infection, *Il18*^−/−^ mice exhibited significantly lower IFNγ levels, higher viral loads, significantly reduced NK cell-mediated cytotoxicity, and increased neutrophil infiltration compared to WT mice, but the humoral and cellular immunity remained unaffected [90]. This suggests that IL-18 protects mice from IAV infection by enhancing NK cell cytotoxicity to control virus replication. In contrast, another study demonstrated that IL-18 was not required for IFNγ production and mouse survival, and that IL-18 hindered virus clearance and weight recovery in survived mice [53].

Compared to WT mice, *Il1r1*^−/−^ mice exhibited increased mortality, reduced neutrophil infiltration, weakened IgM responses, impaired activation and migration of CD4^+^ T cells into the lungs, as well as higher viral loads [91]. Meanwhile, *Il1b*^−/−^ mice showed higher viral antigen expression in the lungs [92]. Coincidentally, another article demonstrated that IL-1β promoted virus clearance by recruiting neutrophils [14]. Moreover, IL-1β can also contribute to the initiation of the adaptive immune response [93], which provides a more specific and long-lasting defense against the virus. Taken together, IL-1β suppresses the virus replication during influenza virus infection.

#### 4.2.5. The Role of Pyroptosis in Influenza Virus Infection

Pyroptosis is a pro-inflammatory form of programmed cell death characterized by the formation of membrane pores, which leads to the release of pro-inflammatory cytokines and damage-associated molecular patterns (DAMPs) [66,94,95]. This cell death pathway plays a crucial role in the host defense against bacterial and viral infections, including the influenza virus [95]. The activation of inflammasomes, particularly the NLRP3 inflammasome, is a key event in the induction of pyroptosis in response to influenza virus infection. NLRP3 inflammasome activation results in the cleavage and activation of caspase-1, which in turn cleaves gasdermin D (GSDMD) to produce the pore-forming GSDMD N-terminal fragment [96]. This process contributes to the release of pro-inflammatory cytokines such as IL-1β and IL-18, which are involved in the recruitment of immune cells and the amplification of the host response to the virus [97]. It has been reported that the interferon (IFN)-inducible protein Z-DNA binding protein 1 [ZBP1, also known as the DNA-dependent activator of IFN regulatory factors (DAI)], activates the assembly of the NLRP3 inflammasome and the execution of pyroptosis in response to influenza viral infection [98]. ZBP1-dependent inflammasome activation and pyroptosis have been shown to restrict influenza virus replication by eliminating virus-infected cells [98], but may also lead to lung inflammation and pathology [98,99,100,101]. Additionally, the overlap between pyroptosis and the release of IL-1β/IL-18 limits the possibility to discern the cytokine-independent effects of pyroptosis in host defense [102]. Further research is needed to fully understand the molecular mechanisms and crosstalk between these processes, and to develop effective therapies that target pyroptosis and its associated pathways in the context of influenza virus infection and inflammatory diseases.

#### 4.2.6. Additional Molecules Associated with NLRP3 Inflammasome in Influenza Virus Infection

Recent studies have shed light on the role of the new key players of NLRP3 inflammasomes in influenza virus infection. Specifically, DEAD-box helicase 3 X (DDX3X), a host protein, plays a critical role in orchestrating the antiviral innate immune response during IAV infection. DDX3X activates the NLRP3 inflammasome in response to the wild-type (WT) influenza A virus (IAV) that carries NS1, and thereby controls viral spread, clears the infected cells, and promotes lung tissue repair [103]. However, in the absence of NS1, DDX3X promotes the formation of stress granules, which facilitates the efficient activation of type I IFN signaling, and confers antiviral activity [103]. This alternate model is critical to the fight against NS1-mediated immune evasion strategies during IAV infection [103]. Additionally, vimentin, a type III intermediate filament (IF) protein, regulates the activation of the NLRP3 inflammasome [104]. Notably, it may promote IAV-induced acute lung injury (ALI) through interaction with NLRP3 [104]. Macrophage migration inhibitory factor (MIF), a multifunctional protein that serves as a vital regulator of innate immunity, is involved in various inflammatory processes and pathological conditions [105,106,107,108,109]. It has been shown that MIF is required for influenza A peptide PB1-F2-induced NLRP3 inflammasome activation [107]. Furthermore, MIF is required for the interaction between NLRP3 and vimentin, and this vimentin-MIF-NLRP3 interaction faciliates NLRP3 inflammasome assembly [107]. The activator protein-1 (AP-1), composed of proteins belonging to the Jun, Fos, and activating transcription factor protein families, is a dimeric transcription factor downstream of mitogen-activated protein kinase (MAPK) signaling. AP1 has been reported to be involved in various cellular events, including the differentiation, proliferation, survival, apoptosis, and synthesis of immune effector molecules [110,111]. During IAV infection, the AP1 signaling pathway played a dominant role in upregulating pro-IL-1β mRNA induced by influenza A virus (IAV) in THP-1 macrophages [112].

#### 4.2.7. Transition of NLRP3 from Detrimental to Protective Functions during Influenza Virus Infection

During influenza virus infection, the activation of NLRP3 inflammasome is considered an essential component of the host’s antiviral defense mechanism. After viral infection, the formation of inflammasomes facilitates the autocatalytic processing of pro-caspase-1, leading to the cleavage and release of pro-inflammatory cytokines IL-1β and IL-18. The production of these cytokines helps to induce an antiviral immune environment, reducing viral replication and spread [97]. However, the activation of the NLRP3 inflammasome can have detrimental effects during influenza infection. (1) Excessive inflammatory response: When the NLRP3 inflammasome is overactivated, it can lead to an excessive inflammatory response, also known as a “cytokine storm” [113]. This is a situation where the immune system not only attacks the virus, but also damages the host tissues, leading to severe disease or even death. (2) Cell death: Pyroptosis, the inflammatory form of cell death triggered by NLRP3, can damage tissues and lead to the release of more inflammatory signals [114], which can perpetuate the cycle of inflammation and damage. (3) Impaired tissue repair: Excessive inflammation and cell death can impair the body’s ability to repair the damaged lung tissue during influenza infection [115], leading to more prolonged disease and a higher risk of complications. (4) Secondary bacterial infections: Immune responses to the virus can make individuals more susceptible to secondary bacterial infections, especially in the lungs [116]. This is a common complication of severe influenza and is often a contributing factor to influenza-related deaths.

There are several possible ways to transit NLRP3 from detrimental to beneficial during influenza virus infection. (1) Regulating NLRP3 activation: The administration of specific inhibitors to prevent the overactivation of the NLRP3 inflammasome could help to minimize IAV pathogenesis [83]. It is important to maintain a balance where the body’s immune response is able to effectively fight off the virus without causing excessive inflammation and tissue damage. (2) Activating mitophagy: Mitophagy, the process of selective mitochondrial degradation, can potentially regulate NLRP3 inflammasome activation [117]. This could be achieved by promoting mitophagy with certain drugs such as berberine (BBR) [118], which could help alleviate influenza-virus-induced inflammatory lesions [118]. (3) Regulating reactive oxygen species (ROS): ROS are small molecules that can act as signaling messengers in many biological scenarios, including immune responses. However, they can also activate the NLRP3 inflammasome, leading to excessive inflammation [119]. Thus, inhibiting ROS production using the antioxidant Mito-TEMPO could be another potential strategy for managing the response of NLRP3 during influenza virus infection [120]. (4) Modulating cytokine levels: Modulating the levels of interleukin-1β (IL-1β) with the use of IL-1 receptor antagonists could be another potential strategy for managing the response of NLRP3 during influenza virus infection [121]. (5) Regulating autophagy: Autophagy is a process that facilitates the degradation and recycling of cellular components, including pathogens such as viruses. It has been found that autophagy can limit NLRP3 inflammasome activation [122], and thus promote the beneficial effects of NLRP3. Enhancing autophagy through drugs such as sirolimus could potentially provide a way of modulating NLRP3 activation during severe IAV infection [123]. These strategies should be further investigated and verified through rigorous studies, including animal models and clinical trials. It is also important to note that each individual’s response to viral infections and treatments can be influenced by a variety of factors, including genetics and overall health status.

## 5. The Role of Commensal Microbiota in Influenza Virus Infection

The symbiotic microbiota, which consists of trillions of microorganisms, plays a crucial role in regulating various host physiological functions, such as digestion, nutrient metabolism, and immune system development [124,125,126,127,128,129,130,131,132,133,134,135,136,137,138]. Recently, mounting evidence has suggested that the gut microbiota modulates the host’s susceptibility to viral infections and contributes to various immune protective mechanisms against influenza virus infection [18,19,20,88,139,140,141,142,143,144,145]. Neomycin-sensitive bacterial communities continuously provide signals for the transcription of *Il1b* and *Il18*, contributing to pulmonary protective immune responses during influenza virus infection [18,139]. The commensal microbiota persistently stimulates the expression of TLR7, MyD88, IRAK4, TRAF6, and NF-kB in TLR7 signaling pathway, defending the host against influenza virus [20]. Moreover, fecal microbiota transplantation (FMT) has been proposed as a therapeutic intervention for influenza viral infection [146,147,148]. FMT involves the transfer of fecal microbiota from donors to recipients in order to restore or modify gut microbial diversity and function. A recent study on mice demonstrated that FMT improved survival and reduced lung inflammation upon influenza infection [146].

Several mechanisms have been proposed to elucidate how the gut microbiota safeguards against influenza virus infection. Some researchers extensively studied mechanisms involving microbial components, such as lipooligosaccharide (LOS), or the production of microbial metabolites, including short-chain fatty acids (SCFAs), such as butyrate, acetate, and propionate, as well as desaminotyrosine (DAT) [19,51,149,150,151,152,153,154,155]. Recently, one study demonstrated that the outer membrane (OM)-associated LOS of *Bacteroides fragilis* triggers the expression of IFN-β via TLR4-TRIF signaling in colonic cDCs, thereby protecting against vesicular stomatitis virus (VSV) and IAV infections [154]. Microbiota-derived butyrate and propionate aid in mitigating lung immunopathology caused by IAV infection [51], whereas acetate enhances type I IFN (IFN-I) production to foster virus clearance and defend against IAV infection [88]. Furthermore, the microbial metabolite DAT, produced by *Clostridium orbiscindens*, combats IAV infection by amplifying IFN-I signaling [155]. Additionally, IFN-I production induced by microbiota in plasmacytoid dendritic cells (pDCs) boosts anti-pathogen immune responses by regulating a distinct transcriptional, epigenetic, and metabolic baseline state in conventional dendritic cells (cDCs) [156]. Moreover, several studies have explored the relationship between gut microbiota and the efficacy of influenza vaccines [157,158,159,160]. One investigation revealed that the composition of gut microbiota in elderly individuals was significantly correlated with their antibody response to the influenza vaccine [157]. Furthermore, recent studies have probed the potential of employing microbiota that expresses influenza virus antigens as vaccine vectors or microbial metabolites as adjuvants for influenza vaccine [161,162,163,164]. For instance, one study determined that a recombinant *Bordetella pertussis* expressing the influenza virus M2e elicited high titers of specific antibodies in mice [161]. Probiotics, defined as “live microorganisms that, when administered in adequate amounts, confer a health benefit on the host” by International Scientific Association for Probiotics and Prebiotics (ISAPP), are becoming increasingly popular. To date, several bacterial strains acting as probiotics with anti-influenza virus activity have been identified in mouse models, including *Bifidobacterium longum* BB536 [165], *Bifidobacterium pseudolongum* NjM1 [88], and various *Lactobacillus* strains [19]. Both *Bifidobacterium longum* BB536 and *Bifidobacterium pseudolongum* NjM1 significantly alleviate influenza virus infection by inhibiting virus replication [88,165]. Among the *Lactobacillus* strains, four strains have also been implicated in inhibiting virus replication, including *Lactobacillus gasseri* TMC0356 [166], *Lactobacillus brevis* KB290 [167], *Lactobacillus acidophilus* L-92 [168], and *Lactobacillus plantarum* DK119 [169,170,171,172,173]. In addition to probiotics, prebiotics have been investigated for their potential to modulate the gut microbiota and improve host responses to influenza infection. A prebiotic is “a substrate that is selectively utilized by host microorganisms conferring a health benefit”. One study found that supplementation with partially hydrolyzed guar gum as prebiotic could influence the intestinal environment, thereby contributing to a reduced incidence of IAV-associated disease [174]. This suggests that interventions targeting gut microbiota may hold promise for the prevention and treatment of influenza viral infection.

## 6. Interplay between the NLRP3 Inflammasome and Microbiota

The intricate interplay between the NLRP3 inflammasome and gut microbiota is vital in maintaining intestinal homeostasis [175], and disruptions in this interaction can lead to various diseases [176]. Both NLRP6 and NLRP3 inflammasomes negatively regulate the progression of non-alcoholic fatty liver disease (NAFLD) and non-alcoholic steatohepatitis (NASH) through the modulation of the gut microbiota [176]. The remodeled gut microbiota, due to *Nlrp3* deficiency, enhances the defense of WT mice against influenza A virus infection following the microbiota exchange between *Nlrp3*^−/−^ and WT mice [88]. Meanwhile, short-chain fatty acids (SCFAs), such as butyrate and acetate, produced by commensal bacteria, have been demonstrated to regulate NLRP3 inflammasome activation [177,178] or NLRP3 expression [88]. Furthermore, bacterial toxins induce the activation of NLRP3 inflammasome, triggering the production of pro-inflammatory cytokines such as IL-1β and IL-18 [179]. In return, these cytokines are essential for maintaining gut epithelium barrier integrity and supporting commensal microbes [175,180].

## 7. Perspectives for Future Research

Future research should explore the complex regulatory mechanisms and interactions of NLRP3 inflammasome with gut microbiota, cellular processes, and co-infections to develop targeted therapeutic strategies and improve patient outcomes. Given the critical role of the NLRP3 inflammasome and gut microbiota in influenza virus infection, targeting their interaction offers a promising therapeutic approach. For instance, modulation of the NLRP3 inflammasome with small molecules, such as nigericin and MCC950, may hold therapeutic potential in treating influenza virus infection. Additionally, strategies aimed at restoring gut microbiota balance, including dietary fiber, probiotics, prebiotics, and fecal microbiota transplantation (FMT), may help to improve morbidity and mortality caused by influenza virus infections. In particular, identifying specific bacterial strains, as well as their functional genes and metabolites that could regulate NLRP3 inflammasome activation using advanced sequencing methods, such as metagenomics, metatranscriptomics, and metabolomics, may provide a targeted approach to prevent or treat influenza virus infection.

Particular attention should also be paid to the interplay between NLRP3 inflammasome activation and other inflammatory pathways, such as the RIG-I/TLR3 signaling pathway, to understand how these pathways cooperatively regulate antiviral immune responses. Additionally, researchers should investigate the connections between inflammasome activation and cellular processes (such as autophagy and pyroptosis) to provide a complete picture of intracellular signaling networks during influenza virus infection. Targeting key components of the pyroptotic pathway, such as NLRP3, caspase-1, or GSDMD, may provide novel treatment options for severe influenza viral infections.

In addition, researchers should expand the current knowledge on the role of NLRP3 inflammasome in co-infections and secondary bacterial infections during influenza virus infection. These co-infections can result in severe clinical outcomes and often lead to increased morbidity and mortality. Understanding the interplay between NLRP3 inflammasome activation and the host response to co-infections will contribute to the development of more effective treatment strategies.

## 8. Conclusions

The NLRP3 inflammasome is a critical component of the host immune response to influenza virus infection, and plays a pivotal role in orchestrating the intricate interplay between the immune system and the gut microbiota. This dynamic relationship holds immense potential for the development of innovative therapeutic strategies aimed at combating influenza viruses. However, to harness this potential effectively, further research is needed in order to gain a deeper understanding of the delicate balance required to control excessive inflammation while simultaneously preserving host immunity. By unraveling this delicate equilibrium, we could strive towards achieving optimal therapeutic outcomes in the battle against influenza viruses.

## Figures and Tables

**Figure 1 viruses-15-01339-f001:**
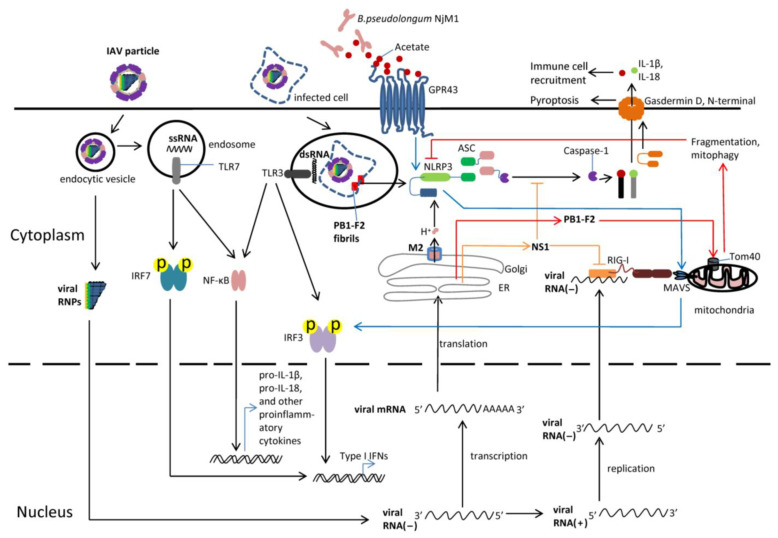
The detection of influenza A virus infection by pattern recognition receptors (PRRs) and NLRP3 inflammasome. Influenza A virus (IAV) infection can be detected by various host sensors that recognize distinct features associated with the infection. Following endocytosis-mediated virus entry, viral ribonucleoproteins (vRNPs) are released into the nucleus for transcription and replication. ssRNA is recognized by TLR7 in the acidified endosomes, and TLR7 signaling triggers the expression of nuclear factor-κB (NF-κB)-dependent pro-IL-1β, pro-IL-18, and other pro-inflammatory cytokines, along with the IRF7 phosphorylation-dependent generation of type I interferons (IFNs). Infected cells are phagocytosed, allowing for double-stranded RNA (dsRNA) recognition by Toll-like receptor 3 (TLR3), which results in the expression of NF-κB-dependent pro-IL-1β, pro-IL-18, and other pro-inflammatory cytokines, as well as type I interferons (IFNs) downstream of IFN-regulatory factor 3 (IRF3). The viral RNA in the cytosol is detected by retinoic acid-inducible gene-I (RIG-I), which activates mitochondrial antiviral signaling protein (MAVS) and induces type I IFNs. Matrix 2 (M2) ion channel activity in the Golgi apparatus stimulates the formation of the NACHT, LRR and PYD domains-containing protein 3 (NLRP3) inflammasome, leading to caspase 1 activation, gasdermin D (GSDMD) cleavage, and the release of cytokines IL-1β and IL-18 via GSDMD N-terminal-formed pores. PB1-F2 fibrils accumulate in the phagosome, which results in the activation of NLRP3 and the release of IL-1β and IL-18. The bacterial strain *B. pseudolongum* NjM1 produces acetate, which enhances viral RNA-triggered MAVS aggregation through GPR43 and NLRP3; such elevated MAVS aggregation promotes subsequent IRF3 activation and IFN-I production. To evade immune responses, the IAV protein PB1-F2 suppresses NLRP3 inflammasome activation by inducing mitochondrial fragmentation and mitophagy, and NS1 inhibits NLRP3 inflammasome activation and blocks the RIG-I signaling pathway. ER, endoplasmic reticulum; GPR43, G-protein-coupled receptor 43; IL, interleukin.

## Data Availability

Not applicable.

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
