# Peer review of "Roles and Mechanisms of NLRP3 in Influenza Viral Infection"

_viruses, 2023, doi:10.3390/v15061339_

Round 1
Reviewer 1 Report
Comments and Suggestions for Authors
It is a comprehensive review discussing the inflammasomes during Influenza infection. The review focused on the followings: a) virus structure and mutation of Influenza virus b) Recognition and innate immune response against the virus either in cytoplasm, endosome or phagosome, and how the virus can evade these strategies. c) Function of NLRP3 inflammasome in influenza virus infection. d) The role of NLRP3 or related proinflammatory cytokines in lung injury repair and pyroptosis. e)Interplay between NLRP3 inflammasome and microbiota and this impact on Influenza infection.
The idea of the review is not new. Several previous published manuscripts described the same concept such as PMID: 28169000,
Major points
1) The authors need to verify what are the new findings/ directions of this review that not previously published. Is it only updated review?
2) The new key players of NLRP3 inflammasomes activation and association with IAV infection are not mentioned or discussed in details , examples of these players : DEAD-box helicase 3X (DDX3X), vimentin and macrophage migration inhibitory factor (MIF).
3) Unless I missed this part, did the authors discussed the new item" Influenza A Virus Infection Activates NLRP3 Inflammasome through Trans-Golgi Network Dispersion (PMID: 35062292).
4) What is the role of Activator protein-1 (AP1) in inflammasome activation during IAV infection
Comments on the Quality of English Language
Moderate language editing
Reviewer 2 Report
Comments and Suggestions for Authors
Generally, the authors presented in detail their topic. The idea of reviewing the contributions of the inflammasome NLRP3 in infection with influenza virus is not new, as it was published previously in some reviews. I appreciate their contribution about the roles of gut microbiota, probiotics, prebiotics and FMT – which should be expanded, given the new available literature. The authors used a lot of references; however, I would suggest using/adding also other very recent titles. My comments/suggestions are listed below:
1. Title: “Anti-influenza Viral Infection” does not make any sense. I would suggest: “Roles and mechanisms of NLRP3 in influenza viral infection” or any other formulation the authors may like, but not “anti-influenza”.
2. Abstract: This one could be better structured, organized, developed, without any reiteration, in order to emphasize more the content of the manuscript. Also, I suggest sentence “Due to…” – lines 17-18, to be inserted after the first sentence (line 16), before “Inflammasomes…”.
3. “Recognition of influenza virus by PRRs”: Line 70 – no need to define again abbreviation “IAV”, as it was done before (line 42).
4. “Function of NLRP3 inflammasome in influenza virus infection”: Please add “s” – “Functions.”
5. “The impact of NLRP3 on host survival and virus clearance”:
a. line 185 – please add “s” to “role”.
b. In the sentence “…demonstrated survival rates of 20% (or 40% according to Thomas et al. 2009), 40% and 30%, respectively [15,16]” – line 187, I suggest writing “rates of 20% [15] (or 40%) [16]”, in order to be precise. Otherwise, Thomas et al has no reference.
6. “Effect of NLRP3 on cytokine production and cell infiltration”: lines 206 – 208 – please insert the missing references, as shown: “Allen et al. [15] and Thomas et al. [16] showed significantly reduced infiltration of monocytes/macrophages and neutrophils in the lungs of Nlrp3-/- and Caspase1-/- mice, while Ichinohe et al. [18] showed more neutrophil infiltration in Nlrp3-/- than in WT mice.”
7. “NLRP3 and lung injury repair”: please revise writing of “Il18”, “Il1r1” etc
8. “The role of commensal microbiota in influenza virus infection”:
a. Please update references, especially [99-105], [106-109] and add more recent ones.
b. Also, when the authors wrote “Several mechanisms…” – line 267, it would be excellent to illustrate these in a figure.
c. Line 268 – After “Some” please insert “researches or “authors”.
d. Line 288 – “To date, several bacterial strains” – please add “acting as probiotics” or “considered probiotics” and also include the definition of probiotics by ISAPP, since later on prebiotics were defined.
e. Lines 294-296 – please use the latest definition of prebiotics by ISAPP, as the one you wrote was old.
f. Line 297 – instead of “prebiotic”, please revise the sentence as “One study found that supplementation with partially hydrolyzed guar gum could influence….”
9. “Concluding remarks”: I would suggest to replace the title with “Perspectives for future research”, as it is well written and insert there all, except for lines 325, 338-340, 347-349 – which could be used in the next paragraph “Conclusion”. Conclusion should be short and crispy and summarize the whole manuscript.
10. References: besides updating them, please write them complete and use a uniform style.
Comments on the Quality of English Language
The English language requires attentive polishing, both for grammar and syntax, also some typos need corrections (e.g. line 175 – “virious”, line 350 – “roel”).
Reviewer 3 Report
Comments and Suggestions for Authors
The current review manuscript aims at discussing the antiviral function of NLRP3 inflammasome against influenza A virus. NLRP3 inflammasome is a double-edged sword to influenza pathogenesis. As discussed in the manuscript (line 170-182), the outcome could depend on the stage of infection in addition to influenza A virus subtypes. To argue NLRP3 as an antiviral protein against influenza A virus, paragraphs 4.2.1 to 4.2.5 were given to discuss the impact and mechanism for NLRP3 in virus clearance, cytokine production and cell infiltration, injury repair, production of antiviral IL-1b and IL-8 and pyroptosis respectively. In addition, by referring to their recent findings in 2023 [Nat Commun. 2023; 14: 642], they have introduced a novel antiviral function of NLRP3 for type I interferon production against influenza virus infection through sensing NjM1-derived acetate.
One major weakness of the current manuscript is that there are too many irrelevant information.
Section 2 and 3 discuss basic biology of influenza A virus and PRR sensing against influenza A virus, which are already discussed topics in some well-known existing review journals and textbooks. Discussion to antigenic drift and shift in 2.2 is irrelevant to the current topic on NLRP3 antiviral effect. For section 3, PRR sensing is slightly more relevant to NLRP3 which is still a member of PRRs. However, as the current review focus on NLRP3, it will be more concise to narrow down the discussion to NLRP3 and its related sensing pathways but put less effort to other sensing pathways such as those discussed in 3.1 to 3.3. NLRP3 was barely found to be discussed in 3.1 to 3.3.
Discussion of gut microbiota to antiviral function of NLRP3 in section 5 and 6 is novel, especially it is relevant to authors’ major recent finding and specialty [Nat Commun. 2023; 14: 642]. Nevertheless, NLRP3 is barely found in section 5’s discussion. Most discussion was given to type I interferon production and viral clearance induced by microbiota-derived molecules. The correlation of NLRP3 might not be found except IL-1b and IL-18. Line 267 to 308 only discuss the beneficial effect of microbiota against influenza disease but did not discuss any relevance to NLRP3 inflammasome. Line 302 to 308 discussion to Phage is more irrelevant to NLRP3 inflammasome. But until section 6, discussion become relevant to NLRP3.
Resolving the irrelevant materials in the manuscript is required to make it better for publication as a review journal. Otherwise, it is difficult for audience to read and understand whether antiviral function of NLRP3 against influenza A virus is properly discussed and argued.
Finally, back to the discussion to the antiviral function of NLRP3 inflammasome against influenza A virus, the discussion is limiting and not satisfactory. Authors did not elaborate how NLRP3 switches role from detrimental to protective. Line 175 mentioned "stage of infection". What does stage means? What factors determine the protective role of NLRP3? How does NLRP3 inflammasome execute its protective role? It will be great if authors consider to discuss these topics which support the argument for the protective role of NLRP3 inflammasome against influenza A virus infection.
Round 2
Reviewer 1 Report
Comments and Suggestions for Authors
No further comments
Comments on the Quality of English Language
moderate language editing
Reviewer 3 Report
Comments and Suggestions for Authors
Revision is still required for the revised part:
- line 305-323, this paragraph is purely subjective without reference and evidences. Need to provide evidences to support it if it is included in the review.
- line 325-349 is a perspective to control NLRP3 response upon influenza A virus infection. Still, after checking reference 117-126, none of the studies are specific to influenza A virus. Two specific points to note:
a) line 326-331, the use of MCC950 or OTL1177 could prevent the antiviral function NLRP3. Targeting NLPR3 alone might not be easy to switch detrimental NLRP3 inflammasome to be beneficial, except a specific study could be provided proving the point.
b) rapamycin could further promote influenza A virus disease (PMID: 28646236). The suggestion in line 342-346 is not valid.
